# AdaptVision: Efficient Vision-Language Models via Adaptive Visual Acquisition

## Abstract

Vision-Language Models (VLMs) have achieved remarkable success in visual question answering tasks, but their reliance on large numbers of visual tokens introduces significant computational overhead. While existing efficient VLM approaches reduce visual tokens through fixed-ratio compression, they operate passively and lack the ability to adapt to varying task requirements. This motivates a fundamental question: Can VLMs autonomously determine the minimum number of visual tokens required for each sample? Inspired by human active vision mechanisms, we introduce AdaptVision, a novel VLM paradigm that enables adaptive visual token acquisition through a coarse-to-fine approach. Our model initially processes compressed visual tokens from low-resolution images and selectively acquires additional visual information by invoking a bounding box tool to crop key regions when necessary. We train AdaptVision using a reinforcement learning framework that carefully balances accuracy and efficiency. Central to our approach is Decoupled Turn Policy Optimization (DTPO), which decouples the learning objective into two components: (1) tool learning, which optimizes correct tool utilization, and (2) accuracy improvement, which refines the generated responses to improve answer correctness. Based on this formulation, we further decouple advantage estimation by computing separate advantages for tokens associated with each objective. This formulation enables more effective optimization for AdaptVision compared to vanilla GRPO. Comprehensive experiments across multiple VQA benchmarks demonstrate that AdaptVision achieves superior performance while consuming substantially fewer visual tokens than state-of-the-art efficient VLM methods.

## 1 Introduction

Recently, Vision-Language Models (VLMs) (Li et al., 2023a; Bai et al., 2023; Chen et al., 2024b) have achieved significant breakthroughs in general visual question answering (VQA) and diverse practical applications by projecting and adapting visual tokens into large language model (LLM) space (Touvron et al., 2023; Achiam et al., 2023; Zhu et al., 2023; Bai et al., 2023). However, the promising performance of VLMs largely relies on the large amount of vision tokens, inevitably introducing a huge memory and computational overhead when compared to LLMs, particularly for high-resolution images. For instance, a $2048 \times 1024$ image yields 2,678 vision tokens in Qwen2.5-VL (Bai et al., 2025). Therefore, it is crucial to avoid the excessive consumption of visual tokens.

Numerous studies have explored visual token compression to enhance VLM efficiency (Yang et al., 2025a; Chen et al., 2024a; Wen et al., 2024; Shi et al., 2023; He et al., 2024; Jian et al., 2023; Zhang et al., 2024b; Yang et al., 2025b). Existing works can be categorized into two main research directions. The first prunes or merges a fixed number of visual tokens based on predetermined thresholds, according to the importance and similarity of vision tokens (Yang et al., 2025a; Chen et al., 2024a; Zhang et al., 2024b). The second dynamically processes distinct samples, where the system adaptively switches between using 100% vision tokens for OCR-related tasks and 25% vision tokens for simpler tasks by selectively employing quarter-resolution images (Yang et al., 2025b). However, existing efficient VLM paradigms and methods are largely *passive*, as they can only reduce the number of vision tokens by predefined ratios. This leads to a natural question: *Can VLMs adaptively determine the minimum number of vision tokens for each sample according to different scenarios?*

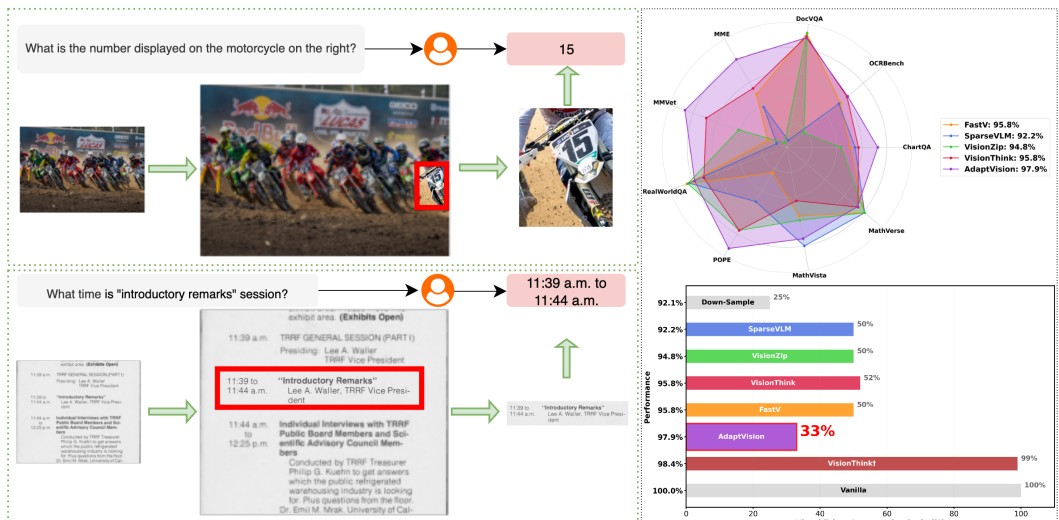

Figure 1: **Our key motivations and AdaptVision performance and efficiency. Left: Coarse-to-fine.** Human visual attention mechanisms first guide the search for question-relevant regions in images, which are then subjected to detailed analysis. **Right:** AdaptVision achieves superior performance with significantly fewer visual tokens than previous efficient VLM methods.

Inspired by human visual processing mechanisms, cognitive neuroscience reveals that our visual system operates not as a passive, uniform sensor but through an active, sequential, and adaptive process known as *active vision* (Findlay & Gilchrist, 2003; Itti et al., 2005). Rather than processing an entire scene at full resolution simultaneously, the human visual system first rapidly captures low-spatial-frequency information (i.e., a coarse overview) to grasp the gist of the scene. It then dynamically allocates attention and eye movements towards salient or task-relevant regions for more detailed, finer-grained analysis (Navon, 1977). This *coarse-to-fine* processing mechanism enables humans to efficiently parse complex visual inputs with minimal cognitive load.

In light of this, we argue that VLMs should similarly adopt an adaptive visual acquisition strategy – one that dynamically adjusts the number of visual tokens based on input content and task, rather than relying on fixed-ratio vision token compression. Such an approach would not only enhance computational efficiency but also reflect a more biologically plausible and intelligent form of visual processing. Fig. 1 provides an illustrative example.

In this paper, we propose AdaptVision, a novel and efficient VLM paradigm that leverages the model's inherent reasoning capabilities. Our model initially processes compressed visual tokens from low-resolution images and adaptively acquires additional visual tokens by invoking a bounding box tool to crop key regions from the original high-resolution image when necessary. The model is trained via reinforcement learning to autonomously decide when to invoke the tool and which region to crop. This is guided by a carefully designed reward function that balances accuracy and efficiency, incentivizing the model to achieve high performance with minimal visual token consumption.

However, training this dual-objective policy with standard RL algorithms like Group Relative Policy Optimization (GRPO) (Shao et al., 2024) presents two key challenges: (1) *Ambiguous credit assignment*: Vanilla GRPO assigns a single sequence-level reward to all generated tokens, failing to distinguish the contribution of the decision to request additional visual tokens from that of generating the final answer; (2) *Imbalanced optimization*: Since vanilla GRPO normalizes all tokens uniformly in a sequence, it introduces an imbalance: compared to 1-turn direct-answer sequences, 2-turn tool-invoking sequences receive imbalanced gradient signals, causing the latter to be under-optimized.

To address these challenges, we propose Decoupled Turn Policy Optimization (DTPO). First, to mitigate optimization imbalance, we decouple the learning objective into two components based on the functional roles of response tokens: (1) *tool learning*, which encourages correct tool use, and (2) *accuracy improvement*, which refines the generated responses to improve answer correctness. Each

objective is normalized separately to balance learning signals across different tokens. Second, to enable precise credit assignment, we decouple advantage estimation by computing distinct advantages for tokens associated with each objective, encouraging more efficient tool exploration. Experiments on multiple VQA benchmarks demonstrate that AdaptVision achieves superior performance with significantly fewer visual tokens than state-of-the-art efficient VLM methods, as shown in Fig. 1.

In summary, our contributions are:

1. We introduce AdaptVision, a novel and efficient VLM paradigm that enables coarse-to-fine visual reasoning.

2. We propose a Decoupled Turn Policy Optimization (DTPO) algorithm alongside a tailored reward function to enable the effective training of AdaptVision.

3. Extensive evaluation on multiple VQA benchmarks shows that AdaptVision achieves superior performance with substantially reduced visual token consumption compared to existing efficient VLM methods.

## 2 PRELIMINARY

### 2.1 REINFORCEMENT LEARNING FOR LARGE LANGUAGE MODELS

Recent studies (Guo et al., 2025; Jaech et al., 2024) have demonstrated Reinforcement Learning (RL) effectively enhances the reasoning capabilities of large language models (LLMs). Recently, Group Relative Policy Optimization (GRPO) (Shao et al., 2024) has been widely used in LLM reasoning. Given a question $x$, GRPO generates $G$ distinct responses $\{o_i\}_{i=1}^G$ from the current policy $\pi_{\theta_{old}}$ and obtains a group of rewards $\{R_i\}_{i=1}^G$. GRPO optimizes the policy model $\pi_\theta$ by maximizing the following objective:

$$\mathcal{J}_{\text{GRPO}}(\theta) = \mathbb{E}_{x,o_i} \left[ \frac{1}{G} \sum_{i=1}^{G} \left( \frac{1}{N_i} \sum_{t=1}^{N_i} \mathcal{L}_{i,t}(\theta) - \beta \mathbb{D}_{\text{KL}} \left[ \pi_\theta(\cdot|x) \,\|\, \pi_{\text{ref}}(\cdot|x) \right] \right) \right], \quad (1)$$

where $\mathcal{L}_{i,t}(\theta)$ denotes the token-level loss formally given by:

$$\mathcal{L}_{i,t}(\theta) = \min \left( \frac{\pi_\theta(o_{i,t} \mid x, o_{i,<t})}{\pi_{\theta_{\text{old}}}(o_{i,t} \mid x, o_{i,<t})} A_{i,t}, \text{clip} \left( \frac{\pi_\theta(o_{i,t} \mid x, o_{i,<t})}{\pi_{\theta_{\text{old}}}(o_{i,t} \mid x, o_{i,<t})}, 1 - \epsilon, 1 + \epsilon \right) A_{i,t} \right), \quad (2)$$

$$A_{i,t} = \frac{R_i - \text{mean}(\{R_i\}_{i=1}^G)}{\text{std}(\{R_i\}_{i=1}^G)}, \qquad \mathbb{D}_{\text{KL}}(\pi_\theta \| \pi_{\text{ref}}) = \frac{\pi_{\text{ref}}(o_i|q)}{\pi_\theta(o_i|q)} - \log \frac{\pi_{\text{ref}}(o_i|q)}{\pi_\theta(o_i|q)} - 1, \quad (3)$$

where $\mathbb{D}_{\text{KL}}$ is the KL-divergence measure. $\epsilon$ and $\beta$ are hyperparameters. The advantage estimate $A_i$ is computed using a group of rewards $\{R_i\}_{i=1}^G$.

### 2.2 VISION LANGUAGE MODELS

The VLM architectures generally consist of three components: a visual encoder, a modality projector, and a LLM. A commonly used approach for the visual encoder is to employ a pre-trained image encoder like CLIP-VIT (Radford et al., 2021) that converts input images into visual tokens. The modality projector adjusts the size of these visual tokens to match the embedding size of LLM and to achieve semantic alignment, enabling the LLM to process visual data effectively. The LLM then integrates the aligned visual and textual information to generate responses.

Existing works have revealed that the computational complexity of VLM is strongly influenced by the sequence length (Yang et al., 2025a), where the sequence length is defined as $n = n_{sys} + n_{img} + n_{question}$. In typical VLM tasks, the number of vision tokens $n_{img}$ is often much larger than the other two, sometimes by a factor of 20. Therefore, reducing the number of vision tokens is the key for improving the efficiency of VLMs.

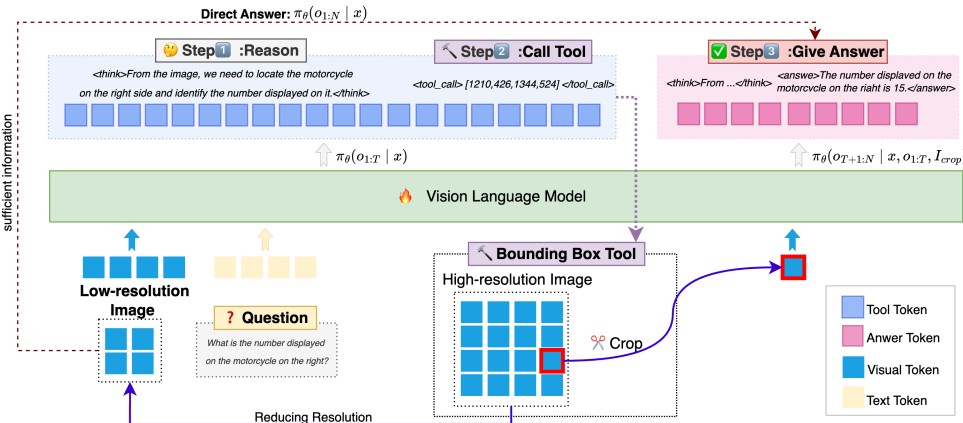

Figure 2: **FrameWork of AdaptVision.** AdaptVision first processes a 1/4-resolution image. The model then decides whether to answer directly or invoke the bounding box tool to crop a high-resolution region for further analysis before generating the final answer.

## 3 METHODOLOGY

### 3.1 FRAMEWORK

We aim to develop an efficient VLM that minimizes visual token usage while maintaining high performance by adaptively acquiring visual information based on question and image complexity. As shown in Fig. 2, our method first processes a low-resolution image ($I_{low}$), cutting visual token usage to 25% of the original. The VLM then autonomously decides whether to answer directly or crop key regions ($I_{crop}$) from the high-resolution image for more detail. To equip the VLM with the ability to generate both direct answers and adaptive visual requests, we design specialized prompts (details in Appendix A.1). As shown in Fig. 2, given a low-resolution image $I_{low}$ and the question $q$, the model can output a *direct answer* or invoke a *tool call* using `<tool_call>[x_1, y_1, x_2, y_2]</tool_call>` to obtain $I_{crop}$ before reasoning further and answering.

Although prompting enables different response styles, the VLM lacks a mechanism for deciding which response style is most appropriate for a given input $x = \{x_{sys}, I_{low}, q\}$. We therefore frame this as a reinforcement learning problem to optimize the following policy:

$$\pi_\theta(o|x) = \begin{cases} \pi_\theta(o_{1:N} \mid x), & \text{direct answer,} \\ \pi_\theta(o_{1:T} \mid x)\,\pi_\theta(o_{T+1:N} \mid x, o_{1:T}, I_{crop}), & \text{tool call,} \end{cases} \tag{4}$$

where $N$ is the length of the entire generated sequence. In tool-call responses, $o_{1:T}$ represents *tool tokens* in the first turn, and $o_{T+1:N}$ represents *answer tokens* in the second turn, as illustrated in Fig. 2. Let $n_{low}$ and $n_{crop}$ be the number of visual tokens for $I_{low}$ and $I_{crop}$. $\mathbf{1}_{tool}$ is the indicator for tool-call responses. Thus, the total number of visual tokens for each sample is: $n_{img} = n_{low} + \mathbf{1}_{tool} n_{crop}$. Therefore, to minimize the number of visual tokens $n_{img}$, we aim to learn a policy $\pi_\theta(o \mid x)$ that can: (1) invoke the tool to request additional visual tokens only when necessary, and (2) acquire the minimal additional visual information $I_{crop}$ required to answer the question correctly.

### 3.2 REWARD DESIGN

To learn a policy that can optimally balance efficiency and accuracy, we design a reward function that consists of two parts: (1) an Outcome Reward $\mathcal{R}_{oc}$ that reflects answer correctness, response format adherence and tool call frequency; (2) a Tool Reward $\mathcal{R}_{tool}$ that incentivizes effective tool exploration to enhance coarse-to-fine visual reasoning. The reward function of AdaptVision is:

$$\mathcal{R} = \mathcal{R}_{oc} + \mathcal{R}_{tool}. \tag{5}$$

**Outcome Reward** $\mathcal{R}_{oc}$. The outcome reward is the sum of three components. (1) *Accuracy reward* $\mathcal{R}_{acc}$: Since VQA answers are typically open-ended, we use an LLM as judge to assign

a binary reward (1 for correct, 0 for incorrect) for answer correctness. The judging prompt is in Appendix A.1. (2) *Format reward* $\mathcal{R}_{form}$: To maintain instruction-following capability, we enforce formatting requirements: reasoning in `<think>` tags, answers in `<answer>` tags, and tool calls in `<tool_call>` tags with valid JSON. The format reward is 0.5 for full compliance with all formatting requirements; otherwise, the reward is 0. (3) *Balance reward* $\mathcal{R}_{bal}$: To prevent over-reliance on tool calls, we introduce a balance reward. We apply a 0.1 penalty to correct answers that invoke tool calls. Additionally, to discourage "lucky guesses" (Yang et al., 2025b), we impose a 0.1 penalty on direct answers when the probability of correct response from low-resolution images is low, thereby encouraging appropriate tool usage. The design of this balance reward is as follows:

$$\mathcal{R}_{bal} = \begin{cases} -0.1 \cdot \mathbb{I}(r < \theta) \cdot \mathbb{I}(\mathcal{R}_{acc} = 1), & \text{direct answer,} \\ -0.1 \cdot \mathbb{I}(\mathcal{R}_{acc} = 1), & \text{tool call,} \end{cases} \qquad r = \frac{C_{direct}}{C_{direct} + C_{tool}}, \qquad (6)$$

where $C_{direct}$ and $C_{tool}$ represent the count of correct answers for direct-answer and tool-call responses within a group, respectively. $\mathbb{I}$ is the indicator function. We set $\theta = 0.2$ in this paper.

**Tool Reward** $\mathcal{R}_{tool}$.  When the model requests additional visual information via a tool call, the cropped region $I_{crop}$ must be both informative for answering and minimal in area to reduce visual token usage. To achieve this balance, we introduce a tool reward $\mathcal{R}_{tool}$, formulated as follows:

$$\mathcal{R}_{tool} = \mathcal{R}_{crop} - \alpha \cdot \mathcal{R}_{area}, \qquad (7)$$

where $\mathcal{R}_{crop}$ evaluates the correctness of the cropped region, $\mathcal{R}_{area}$ denotes its relative area ratio, and $\alpha$ is a hyperparameter balancing the two terms. In this paper we set $\alpha = 2$. (1) The *crop reward* $\mathcal{R}_{crop}$ is determined by GPT-4o, which evaluates whether the cropped region $I_{crop}$ contains relevant information to answer the question, returning 1 if correct and 0 otherwise. The detailed evaluation prompt is provided in Appendix A.1. (2) The *relative area reward* $\mathcal{R}_{area}$ penalizes oversized bounding boxes that contain irrelevant regions, formulated as follows:

$$\mathcal{R}_{area} = \mathbb{I}(\mathcal{R}_{acc} = 1) \cdot \mathbb{I}(\mathcal{R}_{crop} = 1) \cdot \text{clip}\left(\frac{r_{area}}{\mu_{area}(\mathcal{G}(a))} - 1, 0, 1\right),$$

$$r_{area} = \frac{(x_2 - x_1) \cdot (y_2 - y_1)}{H_{low} \cdot W_{low}}, \qquad (8)$$

where $H_{low}$ and $W_{low}$ denote the height and width of $I_{low}$, and $r_{area}$ is the area ratio of the cropped region. Here, $\mathcal{G}(a)$ denotes a group of responses that yield both correct answers ($\mathcal{R}_{acc} = 1$) and correct cropped regions ($\mathcal{R}_{crop} = 1$), and $\mu_{area}(\mathcal{G}(a))$ is the mean measurement of $r_{area}$ within such a group. This area penalty incentivizes the model to select the smallest possible region that still ensures correctness, thereby minimizing visual token usage while maintaining performance.

## 3.3 Efficient Learning via Decoupled Turn Policy Optimization

Based on our reward design, we initially employ GRPO (Shao et al., 2024) for training. We aim to train a VLM that (1) achieves high answering accuracy and (2) minimizes the number of visual tokens used. However, training such a dual-objective policy with GRPO presents two key challenges.

**Ambiguous credit assignment**  Vanilla GRPO provides a single, sequence-level reward to all generated tokens, failing to distinguish between the contributions of two distinct types of actions – the decision to request additional visual tokens and the generation of the final answer. This ambiguity limits effective exploitation and exploration during policy learning. For instance, when the VLM correctly generates bounding boxes while producing an incorrect answer, the model still receives a positive reward for the answer tokens. This may steer the model towards a suboptimal optimization direction. As we will show in the experiments, the model initially favors direct answers but then rapidly collapses to excessive tool call, resulting in an unstable training process.

**Imbalanced optimization**  As defined in Eq. 4, the policy model generates either a one-turn or two-turn responses for each sample. Depending on their functional roles, the generated tokens can be categorized into two types: *Tool Tokens* and *Answer Tokens*, as shown in Fig. 2. Accordingly, the

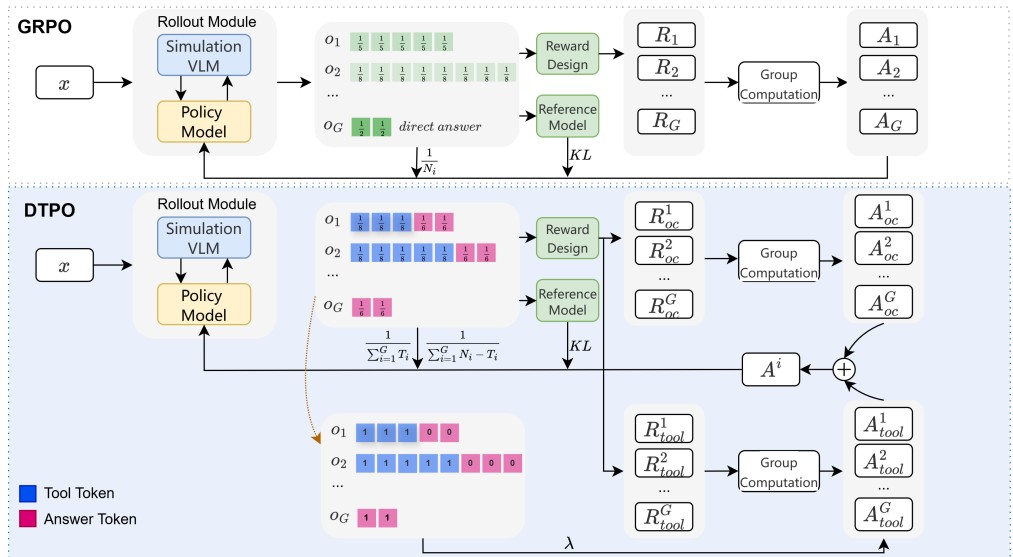

Figure 3: **Demonstration of vanilla GRPO and our DTPO.** Our DTPO (1) decomposes the policy loss by turns to separately optimize tool and answer tokens, and (2) computes distinct advantages for tool and outcome rewards, enabling balanced optimization and precise credit assignment.

original GRPO objective in Eq. 2 can be decomposed into two components:

$$\frac{1}{G}\sum_{t=1}^{G}\frac{1}{N_i}\sum_{t=1}^{N_i}\mathcal{L}_{i,t}(\theta) = \underbrace{\frac{1}{G}\sum_{t=1}^{G}\frac{1}{N_i}\sum_{t=1}^{T_i}\mathcal{L}_{i,t}(\theta)}_{\text{Tool Token}} + \underbrace{\frac{1}{G}\sum_{t=1}^{G}\frac{1}{N_i}\sum_{t=T_i+1}^{N_i}\mathcal{L}_{i,t}(\theta)}_{\text{Answer Token}}, \tag{9}$$

where $T_i$ denotes the number of tool tokens generated in the first turn, and $N_i - T_i$ represents the number of answer tokens in the second turn. If the model answers directly without tool calls, $T_i$ is 0. A closer examination of Eq. 9 reveals an inherent optimization imbalance. In two-turn sequences that invoke tools, the gradient contributions from tool tokens are suppressed by the normalization factors $\frac{1}{N_i}$ and $\frac{1}{G}$, causing tool tokens to be under-optimized compared to answer tokens.

To address these challenges, we propose Decoupled Turn Policy Optimization (DTPO). First, we decouple the policy loss by turns and normalize the contributions of tool and answer tokens separately. This adjustment effectively resolves the under-optimization problem of tool tokens.

$$\mathcal{J}_{\text{DTPO}}(\theta) = \mathbb{E}_{x,o_i}\left[\underbrace{\frac{1}{\sum_{i=1}^{G}T_i}\sum_{i=1}^{G}\sum_{t=1}^{T_i}\mathcal{L}_{i,t}(\theta)}_{\text{Tool Token}} + \underbrace{\frac{1}{\sum_{i=1}^{G}(N_i-T_i)}\sum_{i=1}^{G}\sum_{t=T_i+1}^{N_i}\mathcal{L}_{i,t}(\theta)}_{\text{Answer Token}}\right]. \tag{10}$$

Second, to enable more precise credit assignment, DTPO decouples the advantage estimation by computing distinct advantages for tool and answer tokens, rather than using a single advantage for the entire sequence. Specifically, we compute the advantage for the $t$-th token as follows:

$$A_{i,t} = \begin{cases} A_{oc}^{(i)} + \lambda \cdot A_{tool}^{(i)}, & \text{direct answer,} \\ A_{oc}^{(i)} + \lambda \cdot A_{tool}^{(i)} \cdot \mathbb{I}(1 \le t \le T_i), & \text{tool call,} \end{cases}$$

$$A_{tool}^{(i)} = \frac{\mathcal{R}_{tool}^{(i)} - \text{mean}(\{\mathcal{R}_{tool}^{(i)}\}_{i=1}^{G})}{\text{std}(\{\mathcal{R}_{tool}^{(i)}\}_{i=1}^{G})}, \quad A_{oc}^{(i)} = \frac{\mathcal{R}_{oc}^{(i)} - \text{mean}(\{\mathcal{R}_{oc}^{(i)}\}_{i=1}^{G})}{\text{std}(\{\mathcal{R}_{oc}^{(i)}\}_{i=1}^{G})}, \tag{11}$$

where $\lambda$ is a hyperparameter that trade-offs two advantages. We set $\lambda = 0.3$ in this paper. Fig. 3 compares the design of GRPO and DTPO.

Table 1: **Performance comparison with previous efficient VLM methods.** Vanilla denotes the Qwen2.5-VL-7B-Instruct model. Down-Sample uses a 1/4-resolution image as input to the Vanilla model. "#Token" indicates the visual token consumption ratio relative to the vanilla model across all benchmarks. "Avg." denotes the average performance relative to the vanilla model on all benchmarks.

| Method | ChartQA test | OCRBench test | DocVQA val | MME test | MMVet test | RealWorldQA test | POPE test | MathVista testmini | MathVerse testmini | #Token↓ | Avg.↑ |
|---|---|---|---|---|---|---|---|---|---|---|---|
| *Retain 100% Visual Tokens Across All Benchmarks* | | | | | | | | | | | |
| Vanilla | 79.8 100% | 81.5 100% | 95.1 100% | 2316 100% | 61.6 100% | 68.6 100% | 86.7 100% | 68.2 100% | 46.3 100% | 100% | 100% |
| *Retain 25% Visual Tokens Across All Benchmarks* | | | | | | | | | | | |
| Down-Sample | 62.9 78.8% | 68.8 84.4% | 94.3 99.1% | 2270 98.0% | 54.5 88.5% | 68.8 100.3% | 82.8 95.5% | 62.2 91.2% | 43.1 93.1% | 25% | 92.1% |
| *Retain 50% Visual Tokens Across All Benchmarks* | | | | | | | | | | | |
| SparseVLM | 73.2 91.7% | 75.6 92.7% | 66.8 70.2% | 2282 98.5% | 51.5 83.6% | 68.4 99.7% | 85.5 98.6% | 66.6 97.6% | 45.1 97.4% | 50% | 92.2% |
| FastV | 72.6 91.0% | 75.8 93.0% | 93.6 98.4% | 2308 99.6% | 52.8 85.7% | 68.8 100.3% | 84.7 97.7% | 63.7 93.4% | 45.0 97.2% | 50% | 95.8% |
| VisionZip | 71.5 89.6% | 70.5 86.5% | 93.8 98.6% | 2209 95.4% | 57.0 92.5% | 68.6 100% | 86.3 99.5% | 64.1 93.9% | 45.1 97.4% | 50% | 94.8% |
| *Dynamic Methods* | | | | | | | | | | | |
| VisionThink | 73.6 92.2% | 76.8 94.2% | 92.9 97.7% | 2320 100.2% | 61.7 100.2% | 65.6 95.6% | 86.3 99.5% | 62.2 91.2% | 42.5 91.8% | 52% | 95.8% |
| VisionThink[†] | 73.88 92.6% | 80.8 99.1% | 93.7 98.5% | 2392 103.3% | 60.18 97.7% | 68.37 99.7% | 86.69 100.0% | 65.7 96.3% | 45.68 98.7% | 99% | 98.4% |
| AdaptVision | 75.92 95.1% | 76.9 94.4% | 92.6 97.4% | 2379 102.7% | 64.8 105.2% | 67.32 98.1% | 86.8 100.1% | 65.9 96.6% | 42.3 91.4% | **33%** | **97.9%** |

# 4 EXPERIMENT

## 4.1 EVALUATION SETUP

We conduct experiments on several general VQA benchmarks, including ChartQA (Masry et al., 2022), OCRBench (Liu et al., 2024), DocVQA (Mathew et al., 2021), MME (Fu et al., 2024), MMVet (Yu et al., 2023), RealWorldQA (xAI Team, 2024), POPE (Li et al., 2023b), MathVista (Lu et al., 2023), MathVerse (Zhang et al., 2024a). AdaptVision is based on Qwen2.5-VL-7B-Instruct (Bai et al., 2025). We employ veRL (Sheng et al., 2025) framework for RL training. During training, we set the batch size as 512 and the mini-batch size as 32. We drop the KL term during policy optimization. The initial learning rate of the policy model is $1e - 6$. For each prompt, we sample 16 candidate responses using a temperature of 1.0. During inference, we use the vLLM framework and set the temperature to 0. We use training data from Yang et al. (2025b)[1], which contains VQA samples that can be answered directly using low-resolution images, as well as samples that require high-resolution images for accurate answering. See Appendix A.2 for more details.

## 4.2 MAIN RESULTS

We compare AdaptVision with existing vision token compression methods, including FastV (Chen et al., 2024a), SparseVLM (Zhang et al., 2024b), VisionZip (Yang et al., 2025a), and Vision-Think (Yang et al., 2025b). FastV, SparseVLM, and VisionZip are static methods that operate with a pre-defined token retention ratio, while VisionThink and AdaptVision are dynamic methods that vary visual token usage for each sample. For fair comparison, static methods are set to 50% token retention. For VisionThink, we initially used the officially released model[2] but found it consumed substantially more visual tokens than our method, making the comparison unfair. We thus report two versions: "VisionThink[†]" for the released model and "VisionThink" for our reproduction using the public code. We also include the vanilla model (100% tokens, high-resolution) and the down-sample model (25% tokens, 1/4 resolution). Results are shown in Table 1. Compared to previous vision token compression methods, AdaptVision achieves superior average performance across all benchmarks with significantly fewer visual tokens. Compared to the down-sample model, AdaptVision improves accuracy by 5.8% with only 7% more visual tokens, highlighting its effective coarse-to-fine visual reasoning. We also compare AdaptVision with previous methods in terms of inference time. We report the end-to-end inference time for each dataset in Fig. 7 in Appendix A.3. Compared to the

---

[1]https://huggingface.co/datasets/Senqiao/VisionThink-Smart-Train

[2]https://huggingface.co/Senqiao/VisionThink-Efficient

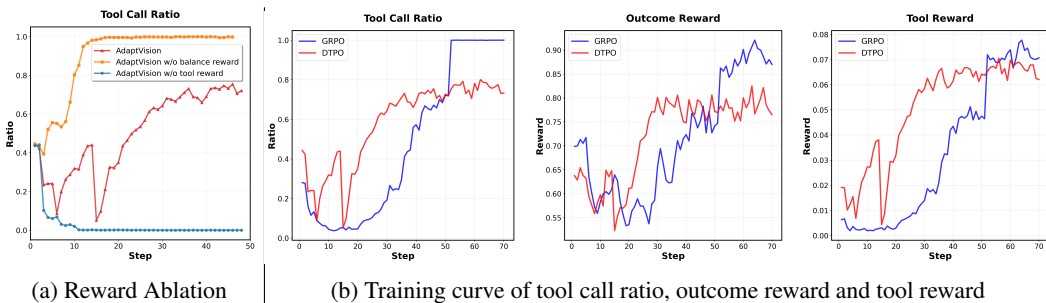

(a) Reward Ablation      (b) Training curve of tool call ratio, outcome reward and tool reward

Figure 4: **Policy-training comparison**: (a) The influence of reward design. (b) GRPO vs. DTPO.

vanilla model and VisionThink[†], AdaptVision demonstrates significantly reduced inference time due to reduced visual token usage. While AdaptVision requires additional tokens for reasoning and tool calls compared to the down-sample model, the increase in inference time remains acceptable.

### 4.3 TRAINING DYNAMICS IN RL

**The Influence of Reward Design**  To investigate the impact of reward design on model behavior, we conduct an ablation study on balance and tool rewards. As shown in Fig. 4a, the absence of the balance reward causes the model to quickly collapse to excessive tool use. This occurs because the tool reward incentivizes correct tool use, which generally improves accuracy as training progresses. Conversely, with balance reward, the VLM learns to adaptively regulate tool usage based on the input. Furthermore, the ablation of the tool reward reveals its necessity for exploration: without it, the model collapses to direct answering and fails to invoke the tool after just 10 training steps. In contrast, with the tool reward, the model successfully explores and leverages the tool to enhance performance.

**GRPO vs. DTPO**  We compare the training processes of GRPO and DTPO in Fig. 4b. GRPO exhibits an unstable training dynamic: During the early training phase, it struggles to optimize either the tool or outcome reward, causing the tool call ratio to drop near zero and limiting exploration. After approximately 20 steps, both rewards and the tool call ratio surge rapidly, shifting the model from direct answering to excessive tool use, eventually collapsing to tool call. This instability stems from GRPO's ambiguous credit assignment and imbalanced optimization. In contrast, DTPO exhibits a stable and efficient optimization process. Both rewards rise steadily from the start, reflecting effective tool use exploration. The model subsequently converges to a reasonable tool call ratio, demonstrating the effectiveness of DTPO. Furthermore, we compare the tool call ratios across different data types. Fig. 10 in Appendix illustrates that our model learns to selectively invoke tools based on task difficulty, while the model trained with GRPO calls tools on all samples, resulting in a 100% tool call ratio.

### 4.4 CASE STUDY

In this section, we present a case study to illustrate the efficient visual reasoning process of AdaptVision. We compare AdaptVision with the vanilla model and the down-sample model. As shown in Fig. 5, the down-sample model, while reducing visual token usage, fails to answer correctly due to insufficient information in the low-resolution image. The vanilla model, using the original high-resolution image, yields a correct answer but at the cost of a large number of visual tokens. In contrast, AdaptVision begins with the low-resolution image, analyzes the question and image, recognizes the informational inadequacy, and then intelligently invokes the tool to crop the most relevant region from the high-resolution image. By acquiring only this essential additional visual information, it produces an accurate answer while minimizing visual token consumption. More cases are provided in Appendix B.

## 5 RELATED WORK

**Vision Language Model with Reasoning.**  Recent advances in reasoning LLMs such as OpenAI's o1 (Jaech et al., 2024) and DeepSeek R1 (Guo et al., 2025) have accelerated the use of RL to enhance

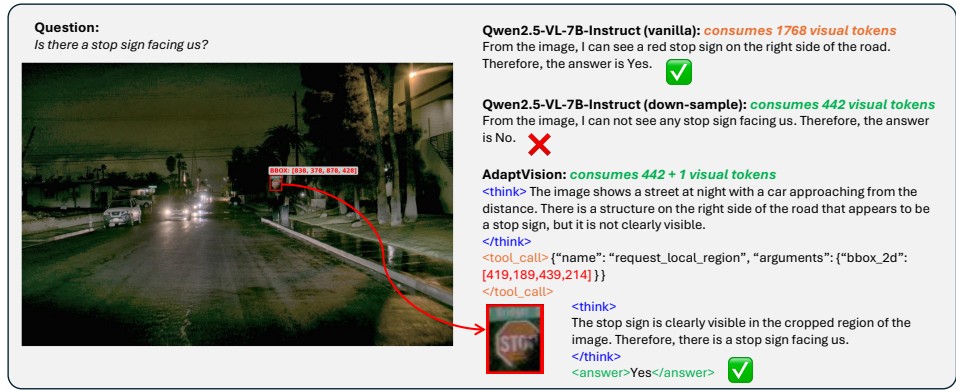

Figure 5: **Case study:** (1) The vanilla model yields a correct answer but consumes a large number of visual tokens; (2) The down-sample model reduces token usage but fails to answer correctly; (3) AdaptVision smartly invokes the tool to produce a correct answer with minimal visual token cost.

reasoning capabilities. This trend has extended to VLMs (Tan et al., 2025; Shen et al., 2025; Liu et al., 2025; Peng et al., 2025), where most work focuses on high-level semantic reasoning like tool use or chain-of-thought explanation. A related direction explores active perception, equipping VLMs with fine-grained control mechanisms (Wu & Xie, 2024; Huang et al., 2025; Su et al., 2025). Recent systems such as DeepEyes (Zheng et al., 2025) and Mini-o3 (Lai et al., 2025) support operations like zoom and crop, improving performance on detailed visual tasks. Unlike these approaches, our method enables the VLM to autonomously determine the minimum number of visual tokens required for a given task, thereby achieving efficient inference while maintaining performance.

**Efficient VLM with Vision Token Compression.** Reducing VLM computational cost by vision token compression has become a popular research topic. Existing methods rely on predefined rules or metrics to compress tokens. For instance, FastV (Chen et al., 2024a) prunes a fixed 50% of tokens based on attention scores after the second layer. PyramidDrop (Xing et al., 2024) proposes progressive token compression to reduce information loss. Other works leverage cross-modal relevance for token selection, such as SparseVLM (Zhang et al., 2024b) and VisionZip (Yang et al., 2025a), which retain semantically relevant visual tokens. A key limitation of these methods is their dependence on a fixed compression ratio, which lacks adaptability across tasks. VisionThink (Yang et al., 2025b) uses RL to decide whether to use a low-resolution or the original image, offering limited adaptability but still restricting the model to coarse-grained decisions. In contrast, our approach enables the VLM to learn coarse-to-fine ability and adaptively determine the minimum number of visual tokens for each task.

## 6 CONCLUSION

In this paper, we present AdaptVision, a novel paradigm that enables VLMs to autonomously determine the minimum number of visual tokens via adaptive, coarse-to-fine visual reasoning. We propose a Decoupled Turn Policy Optimization (DTPO) algorithm, which handles dual-objective policy learning by decoupling the learning objective and advantage estimation. This leads to a more balanced and effective training process than GRPO. Experiments on multiple VQA benchmarks show that AdaptVision achieves superior performance using significantly fewer visual tokens than previous efficient VLM methods. These results advance the development of computationally efficient and biologically inspired VLMs.

Looking ahead, AdaptVision opens new avenues for research in adaptive visual processing and efficient VLM architectures. The framework's ability to dynamically adjust visual token usage based on task requirements suggests promising directions for developing more intelligent and computationally efficient VLM systems.

## ETHICS STATEMENT

Our research is committed to advancing AI responsibly. We exclusively use publicly available datasets (e.g., ChartQA, OCRBench) and open-source base models (Qwen2.5-VL) for all experiments, ensuring transparency and reproducibility. A primary goal of our work is to reduce the computational costs associated with large VLMs, thereby promoting more accessible and sustainable AI. We acknowledge that our model, being built upon a large language model, may inherit societal biases present in its original training data. We encourage further research into the fairness and potential societal impacts of such efficient VLM systems. Additionally, large language models were utilized as a writing aid to improve the clarity and readability of this manuscript.

## REPRODUCIBILITY STATEMENT

To ensure the reproducibility of our research, we provide the following details on our datasets, models, and experiments:

- Experimental Setup: Detailed descriptions of our experimental settings and the baselines used for AdaptVision are provided in Section 4.1 and Appendix A.2.
- Prompt Details: The complete set of prompts used in our method is available in Appendix A.1.
- Dataset and Models: The training dataset and base model are sourced from the open-source community.
- Code Availability: Our source code, configuration files, and experiment scripts will be made publicly available upon publication.

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

# A  ADDITIONAL DETAILS

## A.1  PROMPT DETAILS

AdaptVision utilizes three types of prompts. First, to equip the VLM with basic tool-using capability, we follow the Qwen2.5-VL cookbook (Bai et al., 2025) to design prompts for the bounding box tool (Table 3). Second, since VQA tasks are typically diverse and open-ended, we adopt an LLM-as-judge approach to evaluate answer correctness. As shown in Table 4, following Yang et al. (2025b), we design a judging prompt for GPT-4o to produce binary evaluations (1 for correct, 0 for incorrect). Third, to encourage efficient tool exploration, we prompt GPT-4o to evaluate the relevance of cropped regions, producing a binary reward for region correctness (Table 5).

## A.2  TRAINING AND EVALUATION DETAILS

AdaptVision is based on Qwen2.5-VL-7B-Instruct (Bai et al., 2025). We employ veRL (Sheng et al., 2025) framework for RL training. During training, we set the batch size as 512 with mixed-precision (FP16) training. The mini-batch size is 32. We drop the KL term during policy optimization. For each prompt, we sample 16 candidate responses (i.e., $G = 16$) using a temperature of 1.0. The upper and lower clip ratios are 0.24 and 0.20, respectively. We set the maximum prompt length and the maximum response length as 8192. All experiments were conducted on 4 nodes, each with 8 H20 GPUs. The model was trained for 80 steps, using the AdamW optimizer with a learning rate of $1e-6$, $\beta = (0.9, 0.999)$, and a weight decay of 0.01. During inference, we use the vLLM framework and set the temperature to 0.

## A.3  ADDITIONAL RESULTS

We further compare AdaptVision with previous efficient VLM methods with different visual token retention ratios. As shown in Fig. 6 and Table 2, while the performance of FastV, SparseVLM, and VisionZip degrades with reduced token ratios, AdaptVision maintains superior performance with significantly fewer visual tokens. Fig. 7 shows the comparison of inference time cost. Fig. 10 illustrates that our model learns to selectively invoke tools based on task difficulty, while the model trained with GRPO calls tools on all samples, resulting in a 100% tool call ratio.

# B  QUALITATIVE RESULTS

We provide further case studies to illustrate AdaptVision's adaptive token usage. As shown in Fig. 8, in scenarios where a low-resolution image provides enough information, AdaptVision correctly chooses to answer directly—matching the behavior of the Qwen2.5-VL Down-sample model. Conversely, in cases where detailed visual information is essential (Fig. 9), the Down-sample model often fails due to recognition errors caused by insufficient resolution (e.g., misreading "15" as "75"). Under the same conditions, AdaptVision actively invokes the bounding box tool, accurately localizes informative regions, and produces correct answers with only a marginal increase in visual token consumption relative to the Down-sample model. These examples validate AdaptVision's ability in coarse-to-fine visual reasoning and its capacity to autonomously tailor visual token usage to each input.

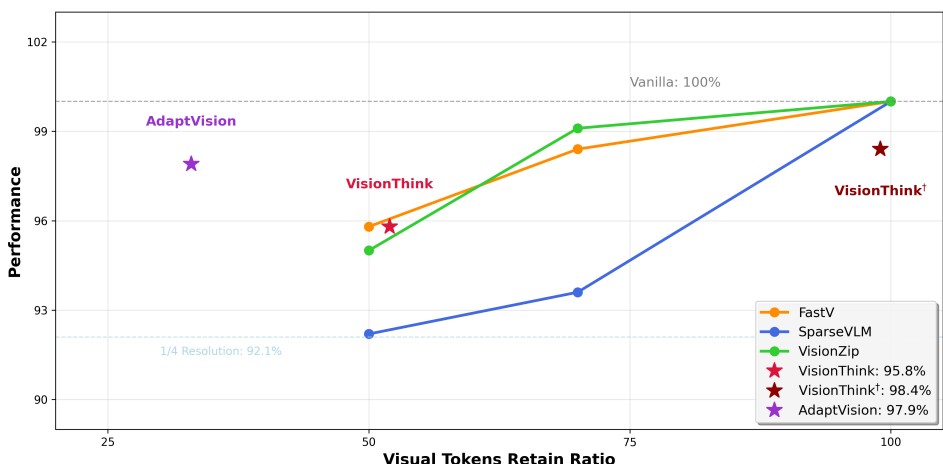

Figure 6: **Performance comparison with different visual token retain ratios.** AdaptVision achieves superior performance with significantly fewer visual tokens than previous efficient VLM methods.

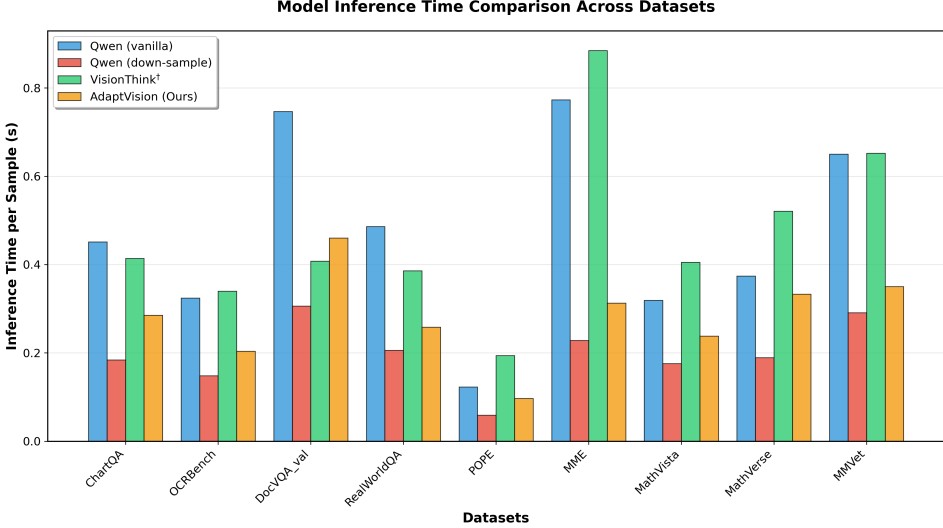

Figure 7: **Comparison of Inference Time.** (1) Compared to the vanilla model and VisionThink[†], AdaptVision demonstrates significantly reduced inference time due to reduced visual token usage. (2) While AdaptVision requires additional generated tokens for reasoning and tool calls compared to the down-sample model, the resulting increase in inference time remains acceptable.

Table 2: **Performance comparison with previous efficient VLM methods.** Vanilla denotes the Qwen2.5-VL-7B-Instruct model. Down-Sample uses a 1/4-resolution image as input to the Vanilla model. "#Token" indicates the visual token consumption ratio relative to the vanilla model across all benchmarks. "Avg." denotes the average performance relative to the vanilla model on all benchmarks. "Method (xx%)" denotes static methods retaining xx% visual tokens.

| Method | ChartQA test | OCRBench test | DocVQA val | MME test | MMVet test | RealWorldQA test | POPE test | MathVista testmini | MathVerse testmini | #Token↓ | Avg.↑ |
|---|---|---|---|---|---|---|---|---|---|---|---|
| *Retain 100% Visual Tokens Across All Benchmarks* | | | | | | | | | | | |
| Vanilla | 79.8 100% | 81.5 100% | 95.1 100% | 2316 100% | 61.6 100% | 68.6 100% | 86.7 100% | 68.2 100% | 46.3 100% | 100% | 100% |
| *Retain 25% Visual Tokens Across All Benchmarks* | | | | | | | | | | | |
| Down-Sample | 62.9 78.8% | 68.8 84.4% | 94.3 99.1% | 2270 98.0% | 54.5 88.5% | 68.8 100.3% | 82.8 95.5% | 62.2 91.2% | 43.1 93.1% | 25% | 92.1% |
| *Retain 50% Visual Tokens Across All Benchmarks* | | | | | | | | | | | |
| SparseVLM (50%) | 73.2 91.7% | 75.6 92.7% | 66.8 70.2% | 2282 98.5% | 51.5 83.6% | 68.4 99.7% | 85.5 98.6% | 66.6 97.6% | 45.1 97.4% | 50% | 92.2% |
| FastV (50%) | 72.6 91.0% | 75.8 93.0% | 93.6 98.4% | 2308 99.6% | 52.8 85.7% | 68.8 100.3% | 84.7 97.7% | 63.7 93.4% | 45.0 97.2% | 50% | 95.8% |
| VisionZip (50%) | 71.5 89.6% | 70.5 86.5% | 93.8 98.6% | 2209 95.4% | 57.0 92.5% | 68.6 100% | 86.3 99.5% | 64.1 93.9% | 45.1 97.4% | 50% | 94.8% |
| *Retain 70% Visual Tokens Across All Benchmarks* | | | | | | | | | | | |
| SparseVLM (70%) | 75.8 94.9% | 79.3 97.3% | 68.7 72.2% | 2276 98.3% | 53.7 87.2% | 68.5 99.8% | 85.4 98.5% | 66.3 97.2% | 45.1 97.4% | 70% | 93.6% |
| FastV (70%) | 71.2 96.7% | 82.2 100.8% | 94.4 99.3% | 2342 101.1% | 56.0 90.9% | 68.6 100% | 85.9 99.1% | 65.9 96.6% | 46.9 101.3% | 70% | 98.4% |
| VisionZip (70%) | 76.8 96.2% | 80.9 99.3% | 94.5 99.4% | 2334 100.8% | 60.0 97.4% | 68.2 99.4% | 86.4 99.7% | 68.9 101.0% | 45.8 98.9% | 70% | 99.1% |
| *Dynamic Methods* | | | | | | | | | | | |
| VisionThink | 73.6 92.2% | 76.8 94.2% | 92.9 97.7% | 2320 100.2% | 61.7 100.2% | 65.6 95.6% | 86.3 99.5% | 62.2 91.2% | 42.5 91.8% | 52% | 95.8% |
| VisionThink[†] | 73.88 92.6% | 80.8 99.1% | 93.7 98.5% | 2392 103.3% | 60.18 97.7% | 68.37 99.7% | 86.69 100.0% | 65.7 96.3% | 45.68 98.7% | 99% | 98.4% |
| AdaptVision | 75.92 95.1% | 76.9 94.4% | 92.6 97.4% | 2379 102.7% | 64.8 105.2% | 67.32 98.1% | 86.8 100.1% | 65.9 96.6% | 42.3 91.4% | **33%** | **97.9%** |

810
811
812
813 Table 3: **Prompt Template for adaptively visual acquisition.** Question will be replaced with the
814 specific question during training and inference.
815

816 *SYSTEM PROMPT:*
817 You are a helpful assistant.
818 # Tools
819 You may call the function tool shown below to assist with the user query.
820 You are provided with the function signature within `<tools></tools>` XML tags:
821 `<tools>`
822 {
823    "type": "function",
824    "function":{
825      "name_for_human": "request_local_region",
826      "name": "request_local_region",
827      "description": "Request a high-resolution local region of the current image and zoom
in",
828         "parameters": {
829         "properties": {
830           "bbox_2d": {
831             "type": "array",
832             "items": {
833                "type": "integer"
834             }
835             "minItems": 4,
836             "maxItems": 4,
837             "description":The bounding box of the region to crop, as [x1, y1, x2, y2], where
(x1, y1) is the top-left corner of the target region and (x2, y2) is the bottom-right corner of
the target region. The bounding box should be in the absolute pixel coordinates of the current
image.",
840           }
841         }
842       "required": ["bbox_2d"],
843       "type": "object",
844     },
   "args_format": "Format the arguments as a JSON object."
845    }
846 }
847 `</tools>`
848 For each function call, return a json object with the function name and the corresponding
849 argument within `<tool_call></tool_call>` XML tags:
850 `<tool_call>`
851 {"name":<function-name>, "arguments":<args-json-object>}
852 `</tool_call>`

853 *USER PROMPT:*
854 Answer the question based on the image provided. You must conduct reasoning within
855 `<think>` and `</think>` first in each of your reasoning steps. You may call ONE func-
856 tion tool per step to help you better solve the problem. Place the function tool within
857 `<tool_call>` and `</tool_call>` at the end of each step to perform a function call.
858 You should continue your reasoning process within `<think>` and `</think>` based on the
859 content returned by the function tool. Once you confirm your final answer, place the final
860 answer inside `<answer>` and `</answer>`. For mathematical or multiple-choice problem,
861 wrap the answer value or choice with `\boxed{}`. Here is the image and question: Question.

862
863

Table 4: **Prompt Template for LLM as Final Answer Judge.** Question, Ground Truth and Prediction are dynamically replaced with the specific question, ground truth and model prediction during evaluation.

---

*SYSTEM PROMPT:*
You are an intelligent chatbot designed for evaluating the correctness of generative outputs for question-answer pairs.
Your task is to compare the predicted answer with the correct answer and determine if they match meaningfully. Here's how you can accomplish the task:
INSTRUCTIONS:
- Focus on the meaningful match between the predicted answer and the correct answer.
- Consider synonyms or paraphrases as valid matches.
- Evaluate the correctness of the prediction compared to the answer.

---

*USER PROMPT:*
I will give you a question related to an image and the following text as inputs:
1. **Question Related to the Image**: Question
2. **Ground Truth Answer**: Ground Truth
3. **Model Predicted Answer**: Prediction
Your task is to evaluate the model's predicted answer against the ground truth answer, based on the context provided by the question related to the image. Consider the following criteria for evaluation:
- **Relevance**: Does the predicted answer directly address the question posed, considering the information provided by the given question?
- **Accuracy**: Compare the predicted answer to the ground truth answer. You need to evaluate from the following two perspectives:
(1) If the ground truth answer is open-ended, consider whether the prediction accurately reflects the information given in the ground truth without introducing factual inaccuracies. If it does, the prediction should be considered correct.
(2) If the ground truth answer is a definitive answer, strictly compare the model's prediction to the actual answer. Pay attention to unit conversions such as length and angle, etc. As long as the results are consistent, the model's prediction should be deemed correct.
**Output Format**:
Your response should include an integer score indicating the correctness of the prediction: 1 for correct and 0 for incorrect. Note that 1 means the model's prediction strictly aligns with the ground truth, while 0 means it does not.
The format should be Score: 0 or 1

---

Table 5: **Prompt Template for Judging the Correctness of Bounding Box.** Question are dynamically replaced with the specific question during evaluation.

---

***SYSTEM PROMPT:***
**Your Role:** You are an AI agent that identifies relevant visual evidence.
**Your Goal:** Determine if an image CROP contains the **primary subject** of a given question.
**Your Golden Rule:** Your main task is to check for **presence**, not completeness. As long as the main object or area the question is asking about is clearly visible in the crop, it is considered relevant.
**Criteria for 'Score: 0' (Strictly Enforced):**
- The core subject of the question is completely absent from the image.
- The image is so blurry or corrupted that the subject is **unrecognizable**.
- The image shows something completely unrelated (e.g. question is about a car, image shows a tree).
**Your Task:**
Now, analyze the user-provided image and question following this exact process. Your response MUST only contain 'Score: 1' or 'Score: 0'.

---

***USER PROMPT:***
Given a question and a cropped image region, answer with 'Score: 1' if the cropped region provide information to answer the question, otherwise answer 'Score: 0'. Question: Question."

---

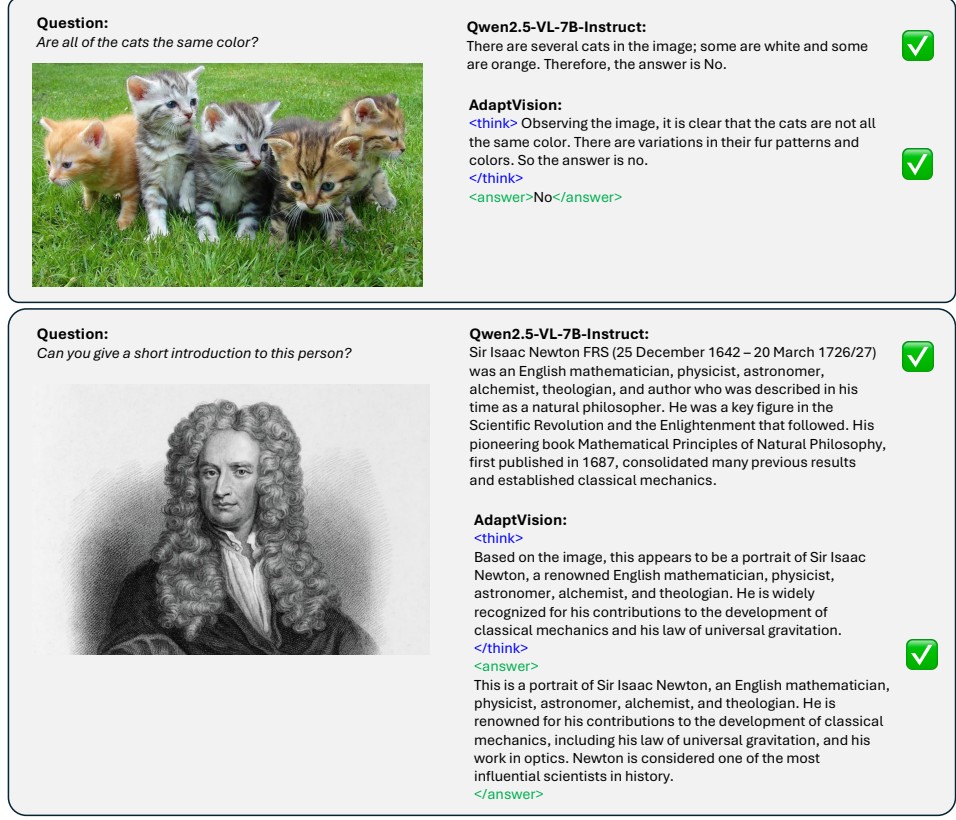

Figure 8: **Case of direct answer in AdaptVision.**

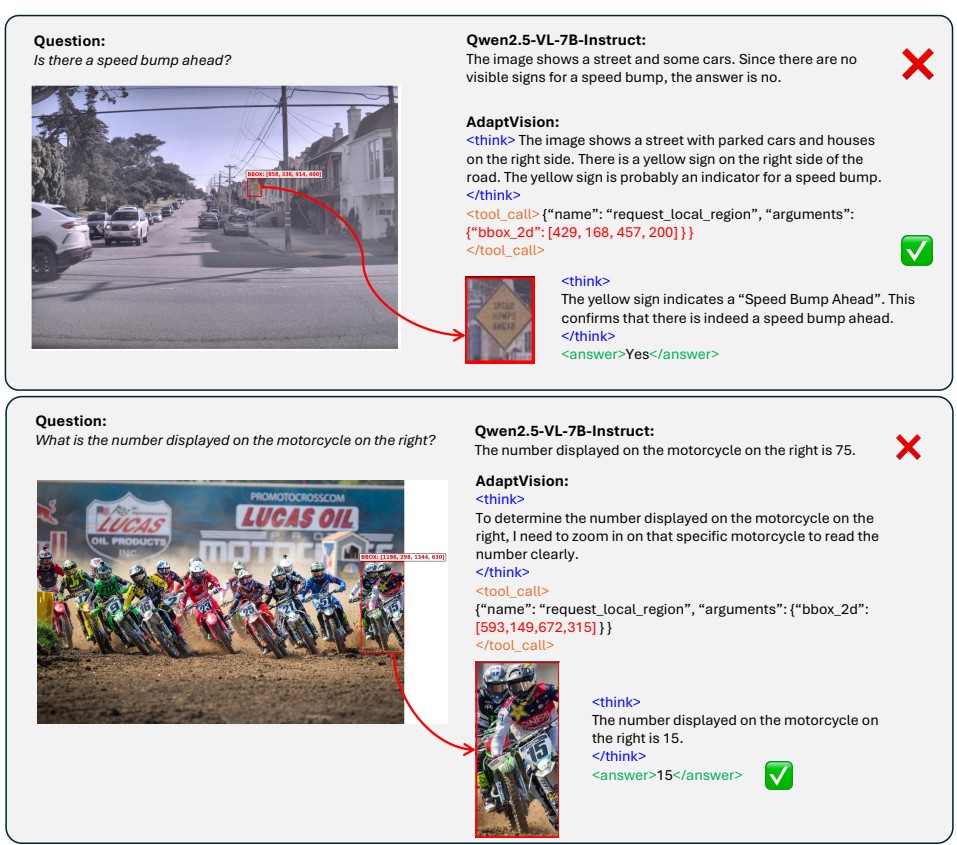

Figure 9: **Case of tool call in AdaptVision.**

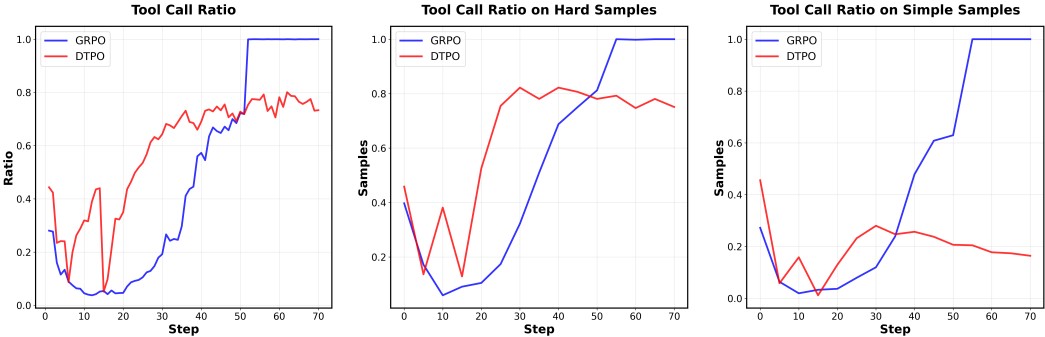

Figure 10: **Training curve of tool call ratio on different types of data.**

