# OpenReview forum: "AdaptVision: Efficient Vision-Language Models via Adaptive Visual Acquisition"
_ICLR.cc/2026/Conference — ICLR 2026 Conference Withdrawn Submission_

### Official Review · Reviewer_7aQJ · 2025-10-16

**Soundness:** 2
**Presentation:** 2
**Contribution:** 2
**Rating:** 2
**Confidence:** 3

**Summary:**

This paper proposes a new RL method, termed DTPO, to teach MLLM to use the bounding box tool, reducing the number of image tokens required during Inference. To achieve the above goals, the authors added a reward item for correct use of the tool, including an evaluation of correctness and requiring the area given by the model to be as close as possible to the size of the target area. Experiments on a series of benchmarks demonstrate the effectiveness of the proposed method.

**Strengths:**

1. The method proposed by the author effectively reduces the usage of tokens while maintaining the original performance of the model.

**Weaknesses:**

1. Lack of necessary explanations for some design.

2. The authors' design lacks the necessary motivation.

Detailed in Questions.

**Questions:**

1. The authors should explain under how models can obtain a higher reward through Eq.8.

2. Lack of direct comparison with GRPO. In the introduction, the authors point out the shortcomings of using GRPO directly, but the experiments lack direct analysis and demonstration of the problems these shortcomings (in line 098-104) may cause. There is also no comparison with the results obtained with GRPO.

3. The authors' reward design lacks clear motivation and comparison. Why is the reward given in this way? What are its advantages and disadvantages compared to the rewad proposed in the VisionThink [a] ?

[a] Senqiao Yang, Junyi Li, Xin Lai, Bei Yu, Hengshuang Zhao, Jiaya Jia. VisionThink: Smart and Efficient Vision Language Model via Reinforcement Learning.

---

### Official Review · Reviewer_fFan · 2025-11-01

**Soundness:** 3
**Presentation:** 3
**Contribution:** 3
**Rating:** 4
**Confidence:** 4

**Summary:**

AdaptVision is a novel, efficient Vision-Language Model (VLM) paradigm that uses an adaptive, coarse-to-fine visual acquisition approach to solve the computational burden of excessive visual tokens. Inspired by how humans actively use their vision , the model first analyzes a low-resolution image and then autonomously decides whether to answer immediately or to call a bounding box tool to crop and acquire minimal, high-resolution visual detail only when necessary for accuracy. To train this dual-goal policy (accuracy and efficiency), the authors introduce Decoupled Turn Policy Optimization (DTPO), a reinforcement learning algorithm that separates advantage computation for tokens that belong to the tool step vs. tokens that belong to the answer step, so long 2-turn sequences don’t get under-trained compared to 1-turn ones. Extensive experiments demonstrate that AdaptVision achieves superior performance across VQA benchmarks while consuming significantly fewer visual tokens, using only a 33% token ratio relative to the vanilla high-resolution model.

**Strengths:**

- Coarse-to-fine visual cropping approach is simple and intuitive.
- Tool call based approach integrates easily with existing LLM ecosystem without breaking standard architectures
- Decoupled RL (DTPO) is simple but effective idea, splitting the advantage into “tool part” vs. “answer part” is a tidy fix to balance multi-turn vs single-turn answers.

**Weaknesses:**

- Extra latency due to potential use of multiple inference turns
- Learned cropping using RL has previously been explored in aesthetic/summarization context (e.g. [a]), and these prior approaches should have been benchmarked as off-the-shelf baselines

[a] Cropper: Vision-Language Model for Image Cropping through In-Context Learning. Seung Hyun Lee et. al. CVPR 2025.

- Some recent works such as ZoomEye[b] have explored learnable zooming/cropping capabilities for MLLMs, these should be reported a baselines.

ZoomEye: Enhancing Multimodal LLMs with Human-Like Zooming Capabilities through Tree-Based Image Exploration. Haozhan Shen et. al. EMNLP 2025.

**Questions:**

- Ablation of reward design would be useful. How sensitive is performance to (i) the weight on token cost, (ii) the bonus for correct answers, (iii) the penalty for unnecessary tool calls?

- Compute the real wall-clock cost. You claim fewer visual tokens overall, but you add a second forward pass on some examples. Can you report: (a) % of examples that triggered a crop, (b) average end-to-end latency vs. a fixed-token baseline, (c) variance over the dataset?

- If trained on “object-centric” images, does it still learn good crops on cluttered UIs, documents, or charts? A cross-domain eval would help.

---

### Official Review · Reviewer_NtA4 · 2025-11-01

**Soundness:** 2
**Presentation:** 2
**Contribution:** 2
**Rating:** 2
**Confidence:** 4

**Summary:**

This paper tackles the challenge of computational overhead in VLMs, which stems from the large number of visual tokens required to process high-resolution images. The authors propose AdaptVision, a new VLM paradigm inspired by human active vision.

Instead of passively processing a fixed number of tokens, AdaptVision operates on a "coarse-to-fine" principle:

It first receives a low-resolution (1/4 size) version of an image, consuming only 25% of the standard visual tokens.

The model then performs an initial reasoning step. Based on this, it makes an autonomous decision:
a.  Answer Directly: If the low-resolution information is sufficient, it generates the answer.
b.  Acquire More Detail: If the information is insufficient (e.g., for a detailed OCR or VQA task), it invokes a "bounding box tool" to request a specific, high-resolution crop of a key region from the original image.

This cropped region provides the necessary detail, and the model then generates the final answer.

The key technical innovation is the training methodology. The authors frame this adaptive decision-making process as an RL problem. They identify critical flaws in standard policy optimization (like GRPO), namely ambiguous credit assignment (is the reward for a good tool call or a good answer?) and imbalanced optimization (direct-answer sequences overpowering tool-call sequences).

**Strengths:**

- The true contribution of this paper is DTPO. While "active perception" or "zoom-in" mechanisms have been explored, the training of such a policy is non-trivial. The authors' analysis of GRPO's failings (ambiguous credit and imbalanced optimization) is insightful, and their solution (decoupling the objective and advantage) is elegant and well-justified.

- The data in Table 1 is impressive. Achieving 97.9% of the vanilla model's performance while using only 33% of the visual tokens is a SOTA-level efficiency trade-off. It clearly outperforms static down-sampling (92.1% perf at 25% tokens) and dynamic methods like VisionThink (95.8% perf at 52% tokens), proving the value of the adaptive, fine-grained cropping approach.

**Weaknesses:**

-  The model is trained on VQA datasets where the task is often "find a specific detail." This "coarse-to-fine-crop" policy is perfectly suited for this. How would this policy fare on tasks requiring holistic scene understanding (e.g., "Describe the overall mood of the image") or complex, multi-object reasoning ("Are the person on the left and the person on the right related?") where a single crop is insufficient? The policy might be overfit to VQA-style problems.

- The two key reward components, $\mathcal{R}_{acc}$ (answer correctness) and $\mathcal{R}_{crop}$ (crop correctness), are determined by an external, proprietary model (GPT-4O). This is a fatal flaw for reproducibility. The agent is not being trained to solve the VQA task; it is being trained to satisfy the preferences of a black-box judge model. Any biases, quirks, or inconsistencies in GPT-4O are directly inherited by the policy. The resulting AdaptVision model is not a generalizable artifact, but a model specifically overfit to the outputs of another model.

- The paper's own related work section (Section 5) cites multiple recent works (e.g., VisionThink, DeepEyes, Mini-O3) that "support operations like zoom and crop" or "decide whether to use a low-resolution or the original image." The paper's claim to novelty seems to rest on the model autonomously determining the minimum tokens, but this is precisely the goal of all dynamic/adaptive methods. The contribution of a tool-based "zoom" is an incremental implementation detail, not a new paradigm as the abstract and introduction repeatedly claim.

**Questions:**

- The paper presents AdaptVision as a "novel VLM paradigm." However, prior work like VisionThink, DeepEyes, and Mini-O3 already implements dynamic resolution switching or "zoom/crop" functionalities. Could you precisely articulate the conceptual novelty of your approach beyond what appears to be an incremental implementation of an existing "active perception" idea?

- How does this approach fundamentally differ from classic two-stage computer vision pipelines (e.g., region proposal + recognition), which also operate on a "coarse-to-fine" principle? Is this not a learned version of a region-of-interest (ROI)-based mechanism?

- The core reward signal for both answer correctness ($\mathcal{R}_{acc}$) and crop quality ($\mathcal{R}_{crop}$) relies on GPT-4O. This introduces a major dependency on an external, proprietary, and non-deterministic model. How can the community reproduce or build upon this work if the reward oracle is a black box?

---

### Official Review · Reviewer_LYZR · 2025-11-01

**Soundness:** 3
**Presentation:** 3
**Contribution:** 3
**Rating:** 6
**Confidence:** 4

**Summary:**

This work studies the VLM by proposing a new agent upon a Decoupled Turn Policy Optimization method. The proposed method learns to adaptively select the region of interest so that to get rid of the uncorrelated visual tokens. A reward shaping is carefully devised to encourage the performance. The proposed method allows improved performance with less visual token consumed.

**Strengths:**

* The proposed method combines the agentic visual tool use with the token compression, which is quite interesting and inspiring.

* The proposed method allows improved performance with less visual token consumed, which could be effective and efficient under some circumstances.

* The proposed method is clear and easy-to-follow.

**Weaknesses:**

* It seems that there lacks failing cases analyze. At what situation the proposed method can fail?

* It seems that there lacks sufficient model discussion, such as oh the configurations of $\alpha$ and $\theta$.

* Despite that this work claims an adaptive visual acquisition method, yet there lacks the statistical analysis on the adaptivity but mainly reports the averaged values.

* How to adaptively adapt the number of selected visual tokens and the performance for different samples?

* Does the proposed method have the risk of the reward hacking? More empirical and quantitative analysis is expected.

**Questions:**

* It seems that the proposed method is limited to perform 1-turn or 2-turn tool calling. How to make sure the region-of-interest can be always captured?

* The proposed method heavily rely on the MLLM to provide the proper region of interest to facilitate the cropping. What might be the performance of the MLLM on choosing the proper regions of interest? Quantitative analysis is expected.

* Accordingly to Fig.4, outcome reward and the tool reward are at the same data range, more interpretations on this is expected. It seems that there is no machoism that normalize both the rewards to the same level.

---

### Note · Authors · 2025-11-14

I have read and agree with the venue's withdrawal policy on behalf of myself and my co-authors.